# When CBCT Looks Borderline and Standard Radiology Is Inconclusive: Should We Plate or Should We Wait?

**DOI:** 10.3390/diagnostics15243140

**Published:** 2025-12-10

**Authors:** Ömer Uranbey, Ece Gülbağ, Büşra Ekinci, Angela Rosa Caso, Jan Nienartowicz, Krzysztof Żak, Kamil Nelke

**Affiliations:** 1Department of Oral and Maxillofacial Surgery, Faculty of Dentistry, Aydın Adnan Menderes University, Aydın 09100, Türkiye; gulbagece35@gmail.com; 2Department of Medical Pathology, Faculty of Medicine, Aydın Adnan Menderes University, Aydın 09100, Türkiye; 3Department of Oral and Maxillo-Facial Surgery, University of Siena, Viale Aldo Moro, 2, 53100 Siena, Italy; ngelarosa.caso@ospedale.perugia.it; 4Department of Maxillo-Facial Surgery, Medical University, 40-027 Wrocław, Poland; 5Academy of Applied Sciences, Health Department, Academy of Silesius in Wałbrzych, Zamkowa 4 Street, 58-300 Wałbrzych, Poland; 6Maxillo-Facial Surgery Ward, EMC Hospital, Pilczycka 144 Street, 54-144 Wrocław, Poland

**Keywords:** mandibular fracture, osteosynthesis, bone healing, prophylactic plating, maxillo-mandibular fixation

## Abstract

The main role of panoramic radiography lies in its rapid screening capability and its ability to detect and identify bone lesions, pathologies, and tooth-bearing structures. Since panoramic radiographs are widely used, they provide a good view of the jaw bones, maxillary sinus, and temporomandibular area. However, their major limitation is the reduced ability to accurately assess bone conditions, particularly in evaluating cortical integrity or identifying subtle, nondisplaced, or greenstick-type fracture lines. Other limitations include the presence of artifacts, image distortion, magnification variability, and high sensitivity to patient and film positioning, all of which can compromise image quality and diagnostic confidence. This 2D imaging method is still used worldwide, especially by dentists; however, this type of radiograph can be unpredictable due to structural superimposition and reduced ability to clearly establish, measure, and verify the precise dimensions, boundaries, and areas occupied by selected lesions. Many patients undergo panoramic imaging to assess possible mandibular fractures after trauma or following the removal of cysts, tumors, or impacted teeth. In most cases, the occurrence of a fracture without displacement can be misjudged, omitted, or underestimated. In such cases, either cone-beam computed tomography is performed or a detailed clinical examination before or during surgery, followed by intraoperative assessment, helps identify a possible fracture line, bone bending, mandibular instability, or the potential need for simultaneous prophylactic plating during dental procedures or the use of maxillomandibular fixation. This paper presents the author’s own experience regarding the limitations of panoramic radiographs in estimating bone condition and detecting fracture lines. Therefore, it is essential to highlight the role of prophylactic (preventive) mandibular plating (PMP) or fixation and to clarify when it should be considered.

**Figure 1 diagnostics-15-03140-f001:**
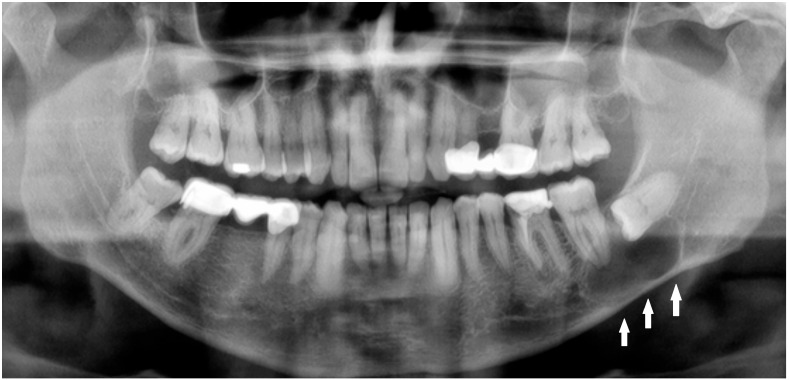
Radiolucent lesions of the mandible often remain unnoticed for years, silently expanding within the bone without any symptoms or visible changes [1]. In certain cases, these deceptively quiescent lesions may weaken the mandibular structure over time and predispose it to unexpected fractures, even after minor trauma [2]. Such an incidental and asymptomatic presentation was observed in the following case. A 50-year-old male patient was referred by a local radiologist following a routine dental panoramic radiograph (DPG) that revealed an incidental finding (Figure 1, white arrows shows the inferior border of the lesion). The patient was systemically healthy, with no history of endocrine disorders such as primary or secondary hyperparathyroidism; metabolic bone diseases such as Paget’s disease; or systemic conditions affecting bone metabolism, including renal osteodystrophy secondary to chronic renal failure, all of which can produce radiolucent lesions in the jaws. Clinically, the patient was asymptomatic, exhibiting no pain, swelling, abscess formation, or facial asymmetry, and no cortical expansion was detected on examination. Radiographic assessment demonstrated that the right mandibular third molar was fully erupted, whereas the left mandibular third molar was completely impacted and retained within the bone. A well-defined, corticated, unilocular pericoronal radiolucency enveloping the crown of the impacted tooth was observed, extending from the enamel–cementum junction, radiographically consistent with a dentigerous cyst. Cone-beam computed tomography (CBCT) confirmed a unilocular lesion with marked buccolingual expansion and cortical thinning in the left posterior mandible but without obvious cortical interruption or a clearly visible fracture line, and the inferior alveolar canal remained corticated and mildly displaced. Nevertheless, because clinical and radiographic findings alone are insufficient for a definitive diagnosis, histopathological evaluation was deemed necessary to confirm the nature of the lesion. Quite often, the extent of a bone lesion may be underestimated, which can lead to an iatrogenic fracture during the removal of a cyst or an impacted tooth. Additionally, a pathological mandibular fracture may occur shortly after surgery. In the present case, the lesion was treated by cyst enucleation and removal of the impacted third molar without prophylactic mandibular plating, as intraoperative mandibular stability appeared satisfactory and no crack propagation or bone bending was palpated. However, in the third postoperative week, the patient developed new-onset pain in the left mandibular region without any reported trauma, and repeat imaging revealed a non-displaced vertical fracture line extending from the distal aspect of the second molar toward the inferior border of the mandible, which was managed conservatively with intermaxillary fixation and close radiological follow-up. In such cases, several considerations should be made: improving diagnostics with CBCT; performing simultaneous preventive or prophylactic mandibular plating (PMP); using other maxillomandibular fixation devices to enhance bone stability, prevent fracture, and promote proper healing; or combining PMP with bone grafting after cyst or tooth removal to significantly reduce the likelihood of mandibular fracture [1,2,3,4]. Three-dimensional assessment of bone condition is fully achievable with CBCT, while the extent of bone loss in the mandible and the size of the three-wall bone defect might influence the use of any bone graft material with or without a PMP. On the other hand, those situations are case-dependent and vary greatly among patients. Secondly, when a standard DPG is used for fracture assessment, whether after typical trauma or following the removal of a large cyst, tumor, or impacted molar, the presence of minimally displaced fracture patterns, very narrow fracture lines, or inconclusive bone findings may lead to failure in identifying a fracture.

**Figure 2 diagnostics-15-03140-f002:**
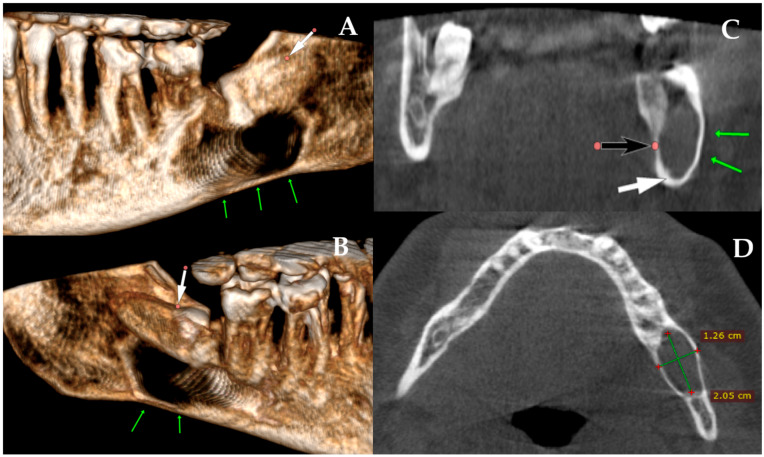
The role of CBCT in contemporary dentistry and oral and maxillofacial surgery remains highly important. Many worrisome or insufficiently diagnosed bone lesions, as well as the extent of bone loss and the risk of fracture, can be more accurately identified and evaluated using CBCT. In the first case presented in Figure 1, a CBCT revealed a well-circumscribed, unilocular, pericoronal radiolucent lesion in the left posterior mandible, enveloping the crown of the impacted mandibular third molar ((**A**) white arrow (decreased bone volume) shows the impacted third molar). The lesion extended anteroposteriorly from the mesial root of the second molar to the anterior aspect of the mandibular ramus (**B**). CBCT demonstrated significant buccolingual expansion with diffuse thinning of both buccal and lingual cortical plates ((**A**–**C**) green arrows showing thinning along the inferior mandibular border), although no cortical perforation was detected. The inferior alveolar canal was displaced inferiorly and slightly lingually ((**C**) white arrow: IAN position; black arrow: lingual plate thinning), but its corticated borders remained intact. The lesion’s internal architecture was homogeneous, without septations or calcifications. The impaction was classified as complete bony impaction, with the cystic lumen closely surrounding the enamel–cementum junction (**B**) (red dots point the arrow). Mild displacement but no resorption of the adjacent second molar roots was observed. The inferior border of the mandible remained continuous, with no radiographic signs suggestive of pathological fracture risk. The axial view additionally demonstrated the degree of buccolingual expansion with dimensional measurements (**D**). Based on CBCT findings, surgery was performed under local anesthesia. A conservative mucoperiosteal flap was elevated to expose the cyst cavity. The lesion was accessed, and complete enucleation with curettage of the cyst lining was carried out. Minimal bone removal was performed to expose the crown of the impacted mandibular third molar and facilitate tooth extraction. After irrigation, the flap was repositioned and sutured primarily. The endodontically failed mandibular left first molar (tooth #36) was also extracted in the same procedure. Although PMP or temporary intermaxillary fixation (IMF) could have been considered, particularly given the degree of buccolingual expansion, both were deferred because intraoperative stability was satisfactory, cortical continuity was maintained, and no crack propagation or bone bending was palpated. Nevertheless, this case demonstrates that even when the mandible appears stable during surgery, marked cortical thinning on CBCT can still increase the likelihood of a delayed postoperative fracture. For this reason, such findings should prompt careful postoperative follow-up or consideration of preventive fixation [5]. In this case, an immediate PMP with a plate or a plate with a bone graft could be used; however, the decision to perform PMP is highly patient-specific and dependent on numerous individual factors.

**Figure 3 diagnostics-15-03140-f003:**
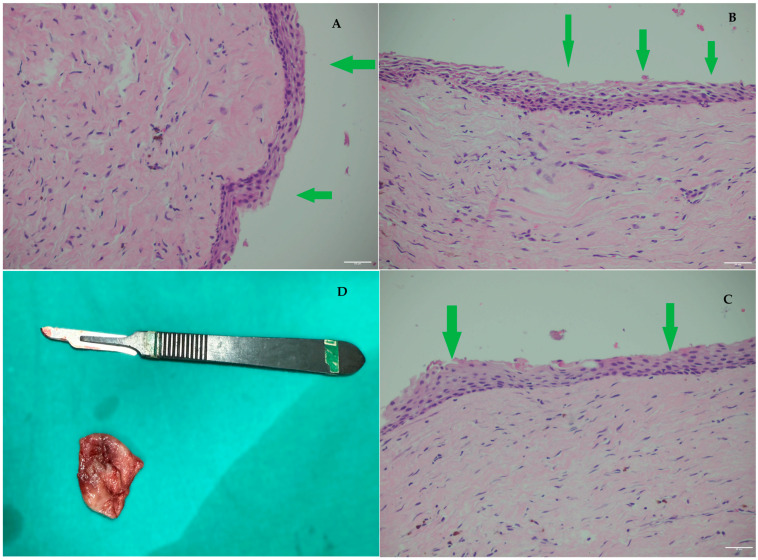
Histopathological examination of the excised cystic lesion revealed a cyst wall composed of loose fibrous connective tissue without significant inflammation, lined by a thin, regular, non-keratinized squamous epithelium ((**A**–**C**) green arrows: thin, continuous non-keratinized epithelial lining without dysplasia) (H&E, ×200 HPF). (**D**): D-lesion diameters measured in CBCT. This evaluation also plays a strategic role in determining the need for prophylactic mandibular plating (PMP). In lesions with atypical radiographic features, rapid growth patterns, or irregular borders, an incisional biopsy may be required before any stabilizing procedure is performed. When a biopsy is scheduled, and a sufficiently large specimen is excised for initial evaluation, this step greatly aids in identifying the lesion type and scheduling further extent of surgery [6]. Establishing whether the lesion is benign, inflammatory, or potentially neoplastic directly influences the surgical approach: conservative cyst enucleation may be sufficient for benign pathology, whereas aggressive or malignant lesions may necessitate wider resection, segmental removal, or planned reconstruction instead of simple plating. Therefore, obtaining a tissue diagnosis either intraoperatively or through a preliminary biopsy ensures that PMP, if indicated, is incorporated appropriately into a treatment plan that reflects both the structural integrity of the mandible and the biological behavior of the lesion. The primary purpose of PMP is to prevent pathological fracture in the operated area during the early postoperative period.

**Figure 4 diagnostics-15-03140-f004:**
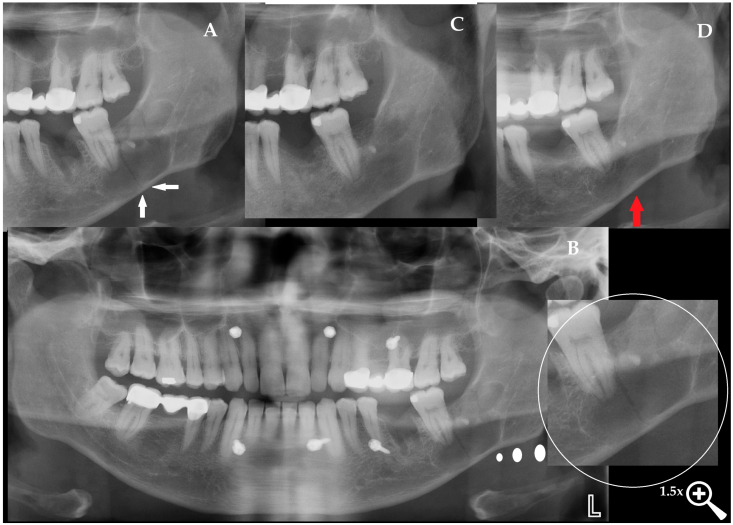
Dentigerous cysts are known to expand slowly and silently over many years, often without producing any symptoms, while progressively thinning the cortical plates to a degree that may compromise mandibular integrity. In such cases, even in the absence of pain, swelling, or functional limitation, postoperative morbidity is inherently increased. The risk of mandibular fracture has been reported both during enucleation and throughout the early postoperative period, particularly in large lesions with marked cortical attenuation, with postoperative fracture rates described in the literature as rare but clinically significant in cases of severe thinning [5]. As presented by Nardi et al., the scope of radiological appearance of mandibular fractures might have some atypical and uncommon patterns; therefore, a DPG should be used for screening, while CBCT or CT should improve the final diagnosis and fracture identification [7]. Even radiologists can misdiagnose any fracture, mostly because sometimes just a simple patient evaluation and good anamnesis can provide more important information. In the third postoperative week, the patient returned unexpectedly, with new-onset pain in the left mandibular region (Figure 4). There was no reported trauma, fall, or identifiable precipitating event. Clinical examination revealed mild tenderness and swelling over the surgical site without occlusal disturbance, prompting immediate radiographic reassessment. The DPG confirmed a linear, vertically oriented radiolucent line extending from the distal aspect of the second molar toward the mandibular base, compatible with a vertical mandibular fracture ((**A**) white arrows indicate the fracture line). CBCT imaging further verified a non-displaced vertical fracture with preserved cortical continuity, absence of cortical step-off, and no interfragmentary gap, confirming a stable fracture pattern without additional pathological findings between the fragments. Given the stable, non-displaced nature of the fracture, intermaxillary fixation (IMF) was selected as the appropriate management strategy to maintain proper alignment of the mandibular segments and ensure controlled healing during the consolidation period. IMF was established, placing mini-screws in both jaws, and elastics were applied after confirming proper occlusion ((**B**) the lesion is highlighted with a white circle). The patient was instructed to maintain a liquid diet and returned one week later for follow-up. At this visit, the elastics were briefly removed to assess mandibular mobility, the oral cavity was irrigated for hygiene, and the fixation was reapplied. Weekly reviews were continued, and by the third week, CBCT imaging demonstrated the first signs of fracture healing with a noticeable reduction in the radiolucent line. The IMF was maintained for a total of four weeks, and at the end of the fourth week, both the mini-screws and elastics were removed. A three-month follow-up DPG confirmed stable healing without complications at the previous screw sites or within the cystectomy area (**C**). The patient was advised to transition to a soft diet, avoid heavy chewing, and return for routine follow-up. At the one-year postoperative evaluation ((**D**) red arrows indicate the healed vertical fracture line), the patient presented with complete functional recovery and no residual symptoms. Radiographic assessment demonstrated complete bone remodeling at the former fracture site, with no trace of the previous cyst cavity or fracture line. When the fracture line is not diagnosed, worrisome granulation tissue formation with or without fistula formation, bone inflammation, soft tissue swelling with irritation, or even an abscess might occur, leading to some potentially life-threatening complications [8].

**Figure 5 diagnostics-15-03140-f005:**
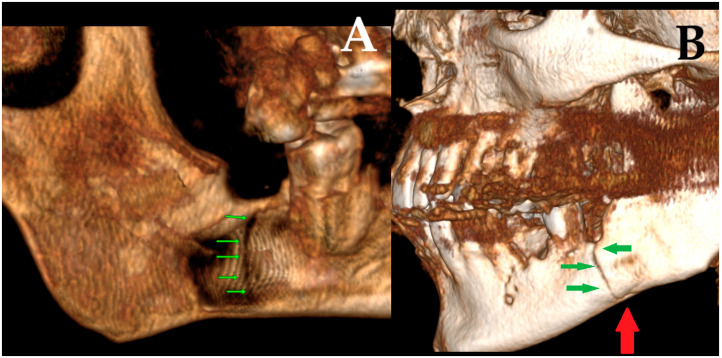
Mandibular fractures occurring after cyst enucleation are uncommon but clinically significant complications, typically associated with extensive cortical thinning and compromised bone integrity. In the present case, a postoperative vertical fracture was detected on tomographic reconstruction ((**A**) green arrows indicating the fracture line in lingual aspect), demonstrating a clear fracture line. The fracture was non-displaced, and the continuity of the cortical margins was preserved, allowing conservative management with intermaxillary fixation using elastics. At the one-month follow-up, a repeat tomographic reconstruction revealed progressive bone healing ((**B**) red arrow shows the healed contour line, trabecular continuity, and stable alignment; green arrows indicate the vertical fracture line and demonstrate progressive osseous healing) of the mandibular border, confirming satisfactory consolidation under elastic guidance. Similar iatrogenic or delayed pathological fractures have been reported in cases involving large cystic lesions or deeply impacted third molars, where chronic expansion leads to progressive cortical attenuation [1,2]. In such scenarios, even minor masticatory forces may precipitate a fracture during the postoperative period. Prophylactic reinforcement with miniplates has therefore been proposed for large or high-risk lesions to prevent pathological fractures [9,10]. Finite element analyses have further demonstrated that rigid internal fixation can restore mandibular stiffness and reduce stress concentrations along weakened borders [4]. More recently, patient-specific titanium plates have been introduced as customized reinforcement tools in extensive cystic or tumor-related defects, enabling optimal adaptation and minimizing postoperative morbidity [3]. Although the fracture in this case healed uneventfully through conservative management, the radiological evidence of significant cortical thinning and the presence of an impacted third molar adjacent to the lesion underscores the potential value of prophylactic plating in comparable cases.

**Figure 6 diagnostics-15-03140-f006:**
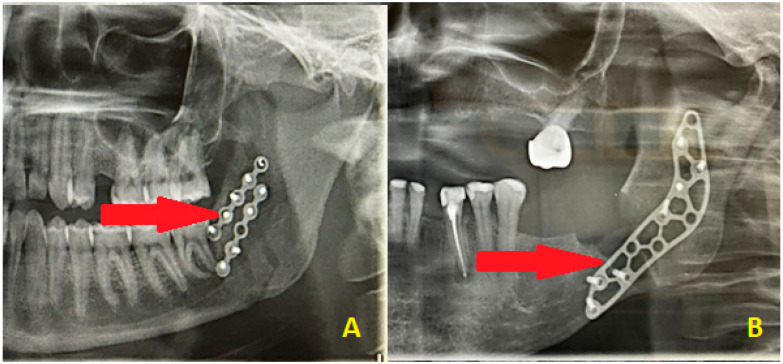
Other limitations of DPGs in assessing the postoperative mandible surgery results might be found in some cases of sagittal mandibular fractures, in which the plating position might not correlate with the fracture line patterns and might suggest an insufficient surgical result (**A**). The second situation (**B**) might be related to a combination of buccal cortical bone loss resulting from surgery on impacted wisdom teeth, which required PMP, while the fracture line can only be assessed intraoperatively and cannot be found on the postoperative radiograph. The red arrows point to the fracture line pattern, which is barely visible on the radiographs. In most cases, a routine DPG is not sufficient to fully evaluate each surgical outcome or assess the presence of a pure and well-defined fracture line; therefore, CT or low-dose CT is recommended. It is worth highlighting that DPG has many limitations. The use of PMP has been reported by a few authors so far, like Murakami et al. and Santos et al., because this approach is not common and strongly linked to individual patient cases, intraoperative features, and the scope of 3D bone loss estimation, which can be measured on CBCT but not on DPG [11,12]. The preventive and prophylactic measures to ensure good mandible bone stability, healing, and immobility are quite important. Other DPG limitations and weak points include only 2D visualization; poor visibility of cortical bone defects; ability to evaluate the position of the condylar head in the glenoid fossa, but not the head angulation; limitations in estimating bone fragment alignment and angulation after surgery; inability to fully assess sagittal fracture lines; inability to fully estimate proper bone alignment; good estimation of proper and symmetrical mandibular ramus height, but ramus slope and rotation not being visible; smaller bone fragments or secondary fractures not being visualized; the lingual cortical bone plate not being visible; non-displaced fractures or greenstick fracture lines being underestimated; and lack of correlation with intra-operative findings during surgery, which could falsely indicate misalignment and inadequate fracture reduction. It is also worth keeping in mind that computed tomography (CT and LDCT) must always be combined with a good patient anamnesis and careful surgical field assessment, sometimes followed by additional radiological studies or even endoscopy or USG to fully assess a typical trauma patient case [13]. On the other hand, where some mandibular surgery is scheduled, the usage of PMP is quite different, but its outcomes include bone stability, avoidance of future pathological fracture, and assurance of bone immobility to promote its adequate healing free of any local inflammation, granulation tissue formation, or other worrisome aspects. That is why CBCT has more to offer than an DPG: an improved fracture evaluation rate; 3D visualization; possibility of evaluation of both cortical and medullary bone; improvements in the condylar head position estimation in the glenoid fossa; a good method for evaluation of bone angulation and proper positioning; all fracture line patterns able to be clearly traced and evaluated; full postoperative bone alignment able to be investigated; improvements in the mandibular ramus evaluation in detail (bone structure, fragments angulation, rotation, height, and width); good visualization of all bone fragments along with the secondary fracture lines; improvements in the lingual cortical plate condition; reliable diagnosis of incomplete mandibular fractures; more improved diagnostics; and more accuracy when establishing the final bone reposition and bone alignment postoperatively. The limitations of panoramic radiographs in post-operative mandibular fracture reduction assessment (2D vs. 3D) should always be taken into consideration.

**Figure 7 diagnostics-15-03140-f007:**
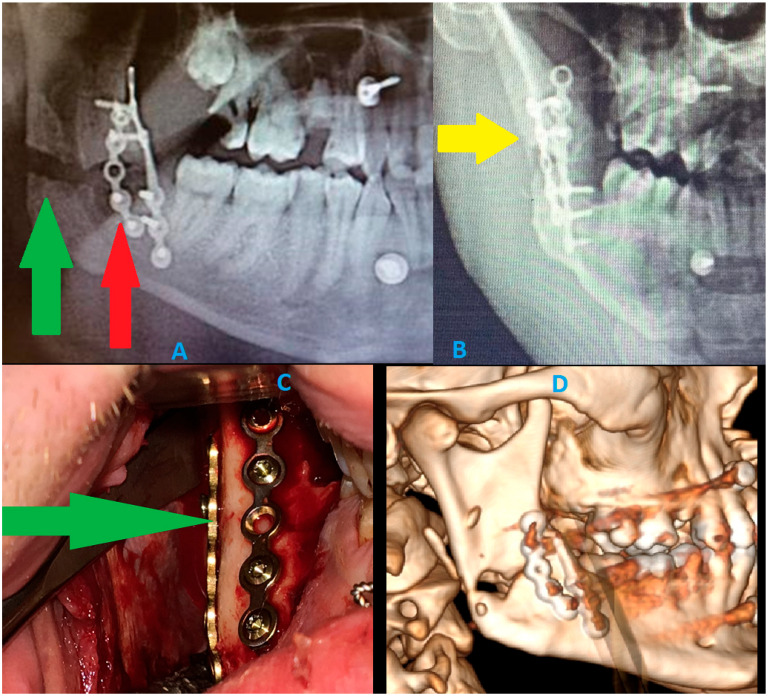
A panoramic radiograph has many limitations. Detecting the presence of a fracture is the first concern; however, a second important aspect is the limited ability to evaluate postoperative fracture healing, bone positioning, and overall stability. These images describe how cortical bone loss can impact the visibility of proper bone relations and their stable outcomes after surgery. These DPG limitations show how intraoperative proper bone reposition and osteosynthesis principles might be misdiagnosed in a routine radiograph [14,15]. A graphical story of the events is shown: (**A**) postoperative routine DPG. The osteosynthesis of both mandibular left basis and right mandibular angle fractures was stabilized (ORIF) with Medartis 2.0 Modus Mandible plating and screw systems (Medartis, Basel, Switzerland) and 4 IMF (intermaxillary fixation 2.0 SpeedTip self-drilling screws 11–14 mm long; Medartis, Basel, Switzerland). Rigid plating systems were used to ensure a good balance between load-bearing and load-sharing forces, related to the forces of the bite, occlusion, and attachment of the masticatory muscles. An additional steel wire was placed in the area of two teeth 34/35 as a Risdon wiring technique to additionally support the fracture line and maintain good occlusion. This control panoramic radiograph (**A**) shows a very good final result and occlusion; however, a large radiolucent area is noticeable at the right mandibular angle. This radiological appearance might suggest bone displacement or mandibular misalignment and a lack of proper reduction. Regarding the bone gap in the right mandibular angle, regardless of the proper condylar head position in the glenoid fossa and proper occlusion, and despite good intraoperative results, the findings were worrisome and might suggest a rotation of two bony fragments, lacking adequate bone contacts for healing, and might require secondary surgery to establish more accurate bone position and angulation. Each clinician should remember that panoramic radiographs (DPGs) have many limitations and should be evaluated with caution [2,3,4,5]. The green arrow points to the bone gap caused by the loss of buccal cortical bone after reposition, while the red arrow points to the loss of buccal cortical bone caused by the removal of impacted wisdom teeth and simultaneous wound debridement. These images (red and green arrows) were suspected to be a bone gap and an inappropriate final surgery result. (**B**) The additional A-P radiograph (anterior–posterior) half mandible projection suggests two bone fragment malalignments, with the occurrence of a radiolucent area suggesting some missing bone fragments (yellow arrow points on a slight crack and bone loss). Because panoramic radiographs are cheap, fast, and easy to make, many surgical results from surgery are not evaluated in postoperative CT scans as often. DPGs could lead to many false diagnoses and misinterpretations of the images; therefore, their usage should be limited. Most commonly, the bone fragment rotations are not noticeable on DPGs; the fracture line can be missed, as well as tilting of some bone fragments can occur. (**C**) The intraoral photograph shows a very good, stable outcome from bone stable reposition and osteosynthesis. Intraoperatively, good bone alignment and bone fragments were achieved. The loosened parts of the bone were removed to improve the repositioning outcome. The loss of some lingual cortical parts of the bone did not affect the repositioning. The green arrow points to a very good fracture reduction line. (**D**) Because the panoramic radiograph was inconclusive and suggested bone malalignment and displacement with bone fragments rotation, which suggested an indication for a secondary revision surgery, a postoperative CT was scheduled. The DPG suggested inappropriate surgery results, which were related to the removal of loosened bone fragments and the necessity to remove all small bone fragments from the fracture line, as well as the missing cortical parts of the bone. The CT examination revealed very good bone contact in the posterior and anterior aspects of the ascending ramus and the posterior part of the mandibular ramus. Despite the missing portion of buccal cortical bone from the ramus, the alignment of bone fragments was satisfactory, with no necessity for any secondary surgery. CT should currently be considered the gold standard for pre- and post-surgery bone evaluation; however, some authors do advise a low-dose CT protocol [3]. In cases of severe trauma, each surgeon has to carefully evaluate each mandibular and craniofacial skeleton trauma to plan the scope and stages of each surgery to restore good facial shape, size, and proper bone alignment. Since the first osteosynthesis methods were described and presented by Champy more than 40 years ago, the proposed protocol for mandibular stable osteosynthesis (ORIF) has continued to be used worldwide [3,5]. Classic routine panoramic radiographs should be evaluated carefully, and a certain degree of inaccuracy may be troublesome for interpretation. CT or low-dose CT should be considered as a more valuable and predictable diagnostic tool in mandibular trauma. From a radiological perspective, a careful DPG evaluation should always be improved with other radiological studies. To summarize, prophylactic miniplate osteosynthesis (PMP) as a preventive measure for the removal of impacted teeth or the removal of any mandibular lesions should be considered on an individual basis. A typical bone plating osteosynthesis used in common mandibular trauma cases, on the other hand, shares certain similarities and differences. One major aspect reveals a major limitation in DPG usage in all presented cases, where the full scope of bone fracture and its boundaries are not always easy to establish. Secondly, when a three-dimensional bone defect shape, volume, and boundaries cannot be evaluated, then some uneventful outcomes may occur for the patient after a mandible lesion surgery (example: impacted teeth, cyst, and tumor removal). Those situations might include bone mobility, bending, instability, or a fracture that might occur sometime after the initial procedure. PMP, with either plates of IMF, offers a better prognosis and enables good healing and bone stability, especially when combined with additional bone grafting materials (xenograft, allograft, and autologous bone) to improve bone volume during the same procedure focused on a specific mandibular lesion [16,17]. DPG has many limitations in fracture evaluation; however, PMP could have a great impact, specifically decreasing pathological fractures of the mandible after some mandibular lesion surgeries. Key Radiologic Teaching Message**:** The presented cases illustrate that panoramic radiography and even CBCT may underestimate the true biomechanical vulnerability of the mandible when large cystic defects cause extensive cortical thinning without an obvious fracture line. A well-defined unilocular pericoronal radiolucency with marked buccolingual expansion and near-complete attenuation of the inferior cortex should be interpreted as a fracture-prone configuration, even if standard radiology appears “borderline” or inconclusive. In such situations, radiologists and surgeons should explicitly report the degree of cortical thinning and consider the mandible at high risk for delayed pathological fracture, thereby prompting discussion of prophylactic mandibular plating or, at a minimum, strict postoperative protection and close radiologic follow-up.

## Data Availability

The data presented in this study are available from the corresponding author on request due to ethical restrictions and protection of patient confidentiality.

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
