# Peer review of "When CBCT Looks Borderline and Standard Radiology Is Inconclusive: Should We Plate or Should We Wait?"

_diagnostics, 2025, doi:10.3390/diagnostics15243140_

Round 1
Reviewer 1 Report
Comments and Suggestions for Authors
This is not a well-written paper and displays numerous careless errors.
Acronyms e.g. OPG appears frequently in advance of their full-extended names. Indeed the correct term for this instrument is dental panoramic radiograph or DPR.
Most of the images pertain to dentigerous cysts and pathological fractures. This should be made clear throughout.
Legends are far to long. Much of their contents should be in the main text.
11 reference out of 17 published in the last 5 years. Good.
Author Response
Response for reviewer 1
- Reviewer comment:
“This is not a well-written paper and displays numerous careless errors. Acronyms e.g. OPG appears frequently in advance of their full-extended names. Indeed the correct term for this instrument is dental panoramic radiograph or DPR. Most of the images pertain to dentigerous cysts and pathological fractures. This should be made clear throughout. Legends are far too long. Much of their contents should be in the main text.”
Response:
We thank the reviewer for this detailed and constructive feedback. We have carefully revised the manuscript in 4 different parts to address all points raised.
Part 1: terminology OPG to DPR
Thank you for pointing this out. We agree that acronyms should only appear after being written in full. But, we now use “dental panoramic radiograph (DPR)” instead of “orthopantomogram (OPG),” and the acronynm is introduced only after the full term. We also reviewed the entire text to ensure that all radiologic terminology is used consistently and accurately.
Part 2: case relations to a dentigerous cyst and a pathological fracture
We appreciate this comment. We have now clearly stated (both in the Abstract and in the corresponding text of Figures 1–5) that the case involves a dentigerous cyst associated with marked cortical thinning and a delayed pathological vertical mandibular fracture. This strengthens our educational purpose of the report and aligns with the reviewer’s suggestion.
Part 3: Figure legends format
Dear Reviewer, thank you for the valuable comment. The special issue SI for diagnostics, MDPI – interesting images is focused on interesting case images followed by radiological data, that presents the case in a special format, very much different than a typical case report. We will try our best to improve the paper as follows.
Part 4: Dear reviewer thank you very much for a very good review, The entire manuscript has been carefully edited to improve clarity, grammar, and flow. We removed repetitive phrasing, improved the precision of radiologic descriptions, and refined the overall structure. These changes have helped make the manuscript clearer and more cohesive.
Reviewer 2 Report
Comments and Suggestions for Authors
The submitted Interesting Images manuscript presents panoramic radiographic findings related to mandibular lesions, emphasizing diagnostic limitations such as reduced sensitivity for subtle fracture lines, cortical interruptions, and greenstick-type injuries. The topic is clinically relevant and the images have potential educational value; however, several revisions are required to meet the standards of this article type.
The panoramic image should be presented with optimal resolution and clearer annotation, ideally with arrows or markers to highlight the subtle radiographic signs. A concise but coherent clinical background is needed to support image interpretation. The radiologic description remains general and would benefit from more precise terminology that clearly emphasizes the main educational point. The manuscript should also articulate the key takeaway message more explicitly, in alignment with the purpose of the Interesting Images format. Additionally, several language issues affect clarity and should be revised, and the ethics statement and author contributions appear incomplete and should be updated according to the journal requirements.
Overall, the submission has potential educational merit, but improvements in image clarity, radiologic description, language quality, and completeness of required statements are necessary before it can be considered for publication.
Author Response
Response for reviewer 2
- Reviewer comment:
“The panoramic image should be presented with optimal resolution and clearer annotation, ideally with arrows or markers to highlight the subtle radiographic signs.”
Response:
We thank the reviewer for this valuable suggestion. In the revised manuscript, the panoramic radiographs has been re-exported in high resolution to improve clarity. Additionally, standardized annotations were added to enhance visualization of the lesion. Specifically, a white circle was incorporated in Figure 4 to highlight the radiolucent area of interest, and a 1.5× magnified inset view was included to provide a clearer depiction of the subtle radiographic changes. Also white arrows added in Figure 1. In Figure 5; the green arrows added to show the osseous healing.
These improvements substantially enhance the interpretability of the image.
Figure 6 has also now been enlarged and optimized to the best achievable quality.
We improved the resolution and clarity of all the figures we were able to re-export from the archive. However, Figures 6 and 7 come from a very old case (over five years ago) that was treated during the COVID-19 period, when our clinical archive system was under heavy strain and many files were stored in compressed formats. We made multiple attempts to retrieve the original high-resolution data, but those files are unfortunately no longer available. Because of this, these two figures could not be upgraded further. All other figures have been successfully improved.
- Reviewer comment:
“A concise but coherent clinical background is needed to support image interpretation.”
Response:
We thank the reviewer for this helpful suggestion. In the revised manuscript, we have added a concise clinical background paragraph to the main text before Figure 1. This paragraph summarizes the patient’s demographic characteristics, relevant medical history, indication for imaging, clinical presentation, radiological progression from OPG to CBCT, and the subsequent development and management of the postoperative mandibular fracture. We believe this addition provides a clearer clinical context for the panoramic radiograph and CBCT images, thereby facilitating image interpretation and understanding of the rationale for (and timing of) prophylactic mandibular plating.
- Reviewer comment:
“The radiologic description remains general and would benefit from more precise terminology that clearly emphasizes the main educational point. The manuscript should also articulate the key takeaway message more explicitly, in alignment with the purpose of the Interesting Images format.”
Response:
We thanks the reviewer for this thoughtful and helpful comment. Now, we have clarified the radiologic descriptions using more precise and clinically meaningful terminology. We now describe the lesion as a welldefined unilocular pericoronal radiolucency with significant buccolingual expansion and thinning of the inferior cortex on CBCT. The postoperative fracture is also detailed more clearly as a nondisplaced vertical fracture line without cortical offset, extending from the distal root of the second molar toward the mandibular inferior border.
We additionally highlight the key contrast that motivated this case: although the panoramic radiograph appeared normal and the CBCT findings seemed “borderline,” the mandible was in fact structurally weakened to a clinically important degree. To make this point more explicit, we have added a short “Key Radiologic Teaching Message” that summarizes the main educational takeaway of this Interesting Images submission namely, that mandibles with marked cortical thinning and large cystic defects should be considered at high risk for fracture even when an obvious fracture line is not visible on OPG or CBCT; this may justify prophylactic plating or, at minimum, stricter postoperative protection and follow up.
- Reviewer comment:
" the ethics statement and author contributions appear incomplete and should be updated according to the journal requirements"
Response
The ethics statement and author contributions information have been completed in the manuscript.
5. Reviewer comment:
“Overall, the submission has potential educational merit, but improvements in image clarity, radiologic description, language quality, and completeness of required statements are necessary before it can be considered for publication.”
Response
Thank you very much for your constructive summary and guidance. We appreciate the detailed feedback and fully understand the areas requiring improvement. We have carefully revised the manuscript to enhance image clarity, refine the radiologic descriptions, improve language quality, and ensure that all required statements are complete and compliant with journal standards.
We sincerely thank you and other the reviewer for the valuable insights provided and hope that the revised version meets the expectations for publication.
Kind regards,
Authors
Round 2
Reviewer 2 Report
Comments and Suggestions for Authors
The authors have adequately addressed all previously raised comments. Image quality and annotations have been improved, a concise clinical background has been added, radiologic terminology has been clarified, and the key educational message is now explicitly stated. Ethical statements and author contributions have also been completed according to journal requirements. Overall, the revised manuscript demonstrates clear improvement and is suitable for further consideration.